# Gonadotropin Releasing Hormone (GnRH) Triggers Neurogenesis in the Hypothalamus of Adult Zebrafish

**DOI:** 10.3390/ijms22115926

**Published:** 2021-05-31

**Authors:** Ricardo Ceriani, Kathleen E. Whitlock

**Affiliations:** Centro Interdisciplinario de Neurociencia de Valparaíso (CINV), Instituto de Neurociencia, Facultad de Ciencias, Universidad de Valparaíso, Valparaíso 2340000, Chile; ricardo_ceriani@hotmail.com

**Keywords:** tanycyte, Pre-optic area (POA), testosterone, BrdU, cytoplasmic Sox2

## Abstract

Recently, it has been shown in adult mammals that the hypothalamus can generate new cells in response to metabolic changes, and tanycytes, putative descendants of radial glia, can give rise to neurons. Previously we have shown in vitro that neurospheres generated from the hypothalamus of adult zebrafish show increased neurogenesis in response to exogenously applied hormones. To determine whether adult zebrafish have a hormone-responsive tanycyte-like population in the hypothalamus, we characterized proliferative domains within this region. Here we show that the parvocellular nucleus of the preoptic region (POA) labels with neurogenic/tanycyte markers vimentin, GFAP/Zrf1, and Sox2, but these cells are generally non-proliferative. In contrast, Sox2+ proliferative cells in the ventral POA did not express vimentin and GFAP/Zrf1. A subset of the Sox2+ cells co-localized with Fezf2:GFP, a transcription factor important for neuroendocrine cell specification. Exogenous treatments of GnRH and testosterone were assayed in vivo. While the testosterone-treated animals showed no significant changes in proliferation, the GnRH-treated animals showed significant increases in the number of BrdU-labeled cells and Sox2+ cells. Thus, cells in the proliferative domains of the zebrafish POA do not express radial glia (tanycyte) markers vimentin and GFAP/Zrf1, and yet, are responsive to exogenously applied GnRH treatment.

## 1. Introduction

Adult neurogenesis plays an important role in controlling brain functions allowing the generation and maturation of new neurons and their integration into neural circuits [1,2]. Neurogenic niches have been identified in specific regions of the adult brain in mammals and fish [3,4,5], including the hypothalamus [6,7]. Unlike mammals, fish produce new neurons along the entire rostrocaudal brain axis throughout life [8] with well-described neurogenic niches in the telencephalon, retina, midbrain, cerebellum, and spinal cord [9]. More recent studies have compared and contrasted the development of mammalian and teleost hypothalamus [10,11], yet few studies have characterized this region in the adult zebrafish.

In mammals, the hypothalamus is characterized by the presence of tanycytes, a specialized type of glial cell that line the wall of the third ventricle in the median eminence, where they contact the cerebrospinal fluid [12,13]. Many studies now suggest that tanycytes have neural stem cell properties and, due to their unique location, may respond to metabolic and/or reproductive cues by modulating hypothalamic neurogenesis [6,12]. Tanycytes can be classified into four groups (α1, α2, β1, and β2) according to their location, morphology, and gene expression [12,14], where only the α2 tanycytes have the neurogenic capacity [15,16,17]. Because the hypothalamus is involved in basic functions, such as controlling metabolism, feeding, body temperature, circadian rhythms, and reproduction [18], the possible role of neurogenesis in controlling these processes is of great interest, especially to develop stem-cell therapies for the treatment of neurological disorders [18,19].

Gonadotropin-releasing hormone (GnRH), secreted by neurons of the preoptic area of the hypothalamus, is essential for controlling puberty and reproduction [20,21]. A decrease in GnRH secretion causes a reproductive syndrome known as Hypogonadotropic Hypogonadism (HH). The clinical manifestation of HH is expressed mainly at puberty, with profound impacts on sexual development, generally resulting in infertility [22]. However, some patients with HH who undergo hormone therapy using testosterone, GnRH, or both show a reversion of HH accompanied by a restoration of pulsatile GnRH release that is maintained after removal of hormone treatment [23,24]. While the mechanism that restores the GnRH level is unknown, the hormone therapy results suggest that testosterone and GnRH could stimulate the differentiation of GnRH neurons in the hypothalamus of adult humans.

Effects of testosterone on neurogenesis have been reported in mammals where androgens can regulate adult neurogenesis in the hippocampus [25] and the subventricular zone (SVZ) of the lateral ventricle [26]. The potential role of GnRH in hypothalamic neurogenesis is apparent in the aging process, where the loss of GnRH function is correlated with decreased neurogenesis [27,28]. Furthermore, our lab has demonstrated that hypothalamic-derived neurospheres of adult zebrafish respond to GnRH with increased neurogenesis, including increases in GnRH neurons in culture [29,30]. These results suggest that testosterone or GnRH stimulates the generation of GnRH neurons from hypothalamic neural progenitors in adult zebrafish. Nevertheless, the effect of GnRH and testosterone in the generation of new GnRH neurons has not been demonstrated in vivo.

Zebrafish are an excellent model system for investigating human diseases, given that approximately 70% of human genes have at least one zebrafish orthologue [31]. They are readily amenable to gene editing [32]; thus, targeted mutagenesis approaches have provided powerful tools to better understand the genetic and developmental basis of human diseases. Because of their rapid development and ability to generate many embryos, zebrafish present a unique opportunity for high throughput drug screening to uncover new drugs for treating human diseases. This study characterizes neurogenesis in the preoptic area (POA) of the adult zebrafish, classifying different neurogenic domains within the region. To elucidate the potential role of testosterone/GnRH in hypothalamic neurogenesis, adult fish were treated with testosterone/GnRH, and effects on proliferation were quantified—revealing that only GnRH treatment significantly increased proliferation in the posterior parvocellular preoptic nucleus of the POA.

## 2. Results

### 2.1. Hypothalamic Neural Progenitors Are Located in the POA

To characterize neural progenitor cells in the pre-optic area (POA) of the adult zebrafish, we first describe the anatomy of the POA, which in fish is defined as located between the medial region of the ventral telencephalon and the optic chiasm (Figure 1A, red box). Forty-six paraffin transverse sections of 5 μmwere obtained from the POA of three fish, and representative sections of the parvocellular preoptic nucleus (PP: PPa0, PPa1, PPa2, PPa3, PPa4.1, PPa4.2, PPp1, and PPp2) were analyzed (Figure 1B–I). The anterior parvocellular preoptic nucleus (PPa) begins at the anterior commissure; the last one is divided into the pars dorsalis (Cantd) and the pars ventralis (Cantv) (Figure 1B,C). These nuclei are localized adjacent to the diencephalic ventricle (DiV) (Figure 1C–G). In the ventral region of the PPa4 sections, cells with fusiform nuclei were found along the wall of the DiV (Figure 1G, boxed, arrow). Cells with fusiform nuclei were no longer apparent in the posterior parvocellular preoptic nucleus (PPp)/suprachiasmatic (SC) (Figure 1H,I) in agreement with Wulliman [33].

Tanycytes, glial-like cells found lining the ventricle in the mammalian hypothalamus are characterized by a variety of markers, including glial fibrillary acidic protein (GFAP) (a standard marker of glial cells), intermediate filament marker vimentin [17], and Sox2 [34]. Thus, to characterize potential alpha-tanycyte-like progenitor cells in the POA, we selected five representative transverse sections and examined the expression of anti-vimentin (Vim), anti-Zrf1 (homologous to GFAP), and anti-Sox2. In PPa1-PPa2 sections sampled from four fish, Vim+/Zrf1+ cells with the morphology of radial glia were observed in the dorsal wall of the DiV (Figure 2A,B, boxed areas). These cells extend their processes into the PPa (Figure 2A’B’, arrows). Additionally, we identified Vim+/Zrf1+ cells in the ventrolateral region of the POA with long processes that extend towards the DiV (Figure 2A,B, asterisks). In PPa3 sections, there were fewer Vim+/Zrf1+ cells lining the wall of the DiV (Figure 2C, boxed area, C’, arrowhead) with Vim+/Zrf1+ cells in the ventrolateral region of the POA (Figure 2C, asterisk). Moreover, a group of Vim+/Zrf1- cells were observed ventrally in the POA, contacting the floor of the DiV with a short process (Figure 2C, green, arrowhead). In posterior sections of the POA (PPa4), few Vim+/Zrf1+ cells were observed (Figure 2D), and cells with fusiform nuclei located in the ventral region were observed (Figure 2D, boxed area, D’, asterisk). In PPp1, Vim+/Zrf1+ cells lining the DiV were observed dorsally in the region of the magnocellular nucleus (PM) and PPp (Figure 2E, boxed area, E’ arrowhead), but not in the region of the suprachiasmatic nucleus (Figure 2E, SC). Although the Vim+/Zrf1+ cells in PPp1 (Figure 2E’ arrow), were similar to those observed in PPa1 and PPa2 sections, these cells extended their processes toward the lateral side of the POA.

In mammals, subsets of tanycytes also express Sox2, a transcription factor essential for neural stem cell differentiation and consistent with the observations that tanycytes can also act as neural stem/progenitor cells. We observed cells immunopositive for Sox2 throughout sections of the PPa/PPp. This expression had a homogeneous distribution along the wall of the DiV (Figure 2F–J; *n* = three fish). In the wall of the DiV, Vim+ cells were also Sox2+ (Figure 2F’–H’ and J, arrow). However, the Vim+ cells are located in the ventral-lateral region were Sox2− (Figure 2F–H, green, arrowheads). In PPa4, the cells with fusiform nuclei were Sox2+ (Figure 2I boxed area, I’, arrow). Thus, the POA contains Vim+/Zrf1+/Sox2+ cells with the morphology of radial glia and ventrally located Sox2+ cells with fusiform nuclei. 

### 2.2. Cytoplasmic Sox2 Cells Express Fezf2:GFP in Ventral Region of the POA

The transcription factor Sox2 is expressed in stem cells and progenitors, as well as differentiated neurons and glia [35]. Previously we described a group of HuC positive neurons with cytoplasmic anti-Sox2 labeling in the parvocellular nuclei [29]. Here we identified Sox2+ cells adjacent to the DiV located in PPp1 sections that had large nuclei (diameter: 6.41 ± 1.2 μm) and labeling in the cytoplasm (Figure 3A; six fish). Because PPp1 corresponds to the neurosecretory preoptic area (NPO), these cells with Sox2+ cytoplasmic labeling may have a neuroendocrine function. We next determined whether the cytoplasmic Sox2+ cells co-localized with markers for neuroendocrine cell specification. In zebrafish, the gene forebrain embryonic zinc finger (*fezf2*) regulates Orthopedia (*Otp*) [36], a transcription factor essential for neuroendocrine cell specification [37]. To determine whether cytoplasmic Sox2+ cells co-localized with Fezf2:GFP, anti-Sox2 antibody labeling was done in *Tg(fezf2:gfp*) adults [38] (*n*= three fish). In the PPp1 sections, two types of cells were GFP+: large cells with low GFP expression (Figure 3B,C,E,F, arrows) and small cells with high GFP expression (Figure 3B,C,E,F, arrowheads). The large cells positive for Fezf2:GFP were also Sox2+ (Figure 3C,F, arrowhead). In contrast, the small cells with high expression of Fezf2:GFP were negative for cytoplasmic Sox2 (Figure 3B,C,E,F, arrow). These results suggest that cytoplasmic Sox2 is found in a group of endocrine cells located in the NPO.

### 2.3. Proliferative Cells in the POA

It is well known that fish generate neurons throughout life [8], yet the neurogenic potential of the hypothalamus is not well described in adult zebrafish. To identify anti-Vim+ potential tanycyte-like precursor cells in the POA, sectioned tissue was double-labeled for anti-Vim and anti-proliferating cell nuclear antigen (PCNA; Figure 4A–E,D’,D’’, three fish). We observed PCNA+ cells lining the DiV in almost all sections (Figure 4A–E, arrow), however these cells were Vim-. The majority of PCNA+ cells, (62%), were located in PPa4 sections (Figure 4D, rectangle; Appendix A), including cells with fusiform nuclei (Figure 4D’–D’’’). To further characterize potential proliferative capacity, sections were double-labeled for anti-Sox2 and anti-PCNA. In all sections, we observed PCNA+/Sox2+ cells (Figure 4F–J, arrow). Similar to what was observed previously (see Figure 4D,D’’,D’’’, arrow), the greatest number of PCNA+ cells were observed in the ventral region (Figure 4F–J, arrow), particularly in the PPa4 (Figure 4I, boxed region, I’–I’’’, arrowhead) where all PCNA+ cells were also Sox2+. Therefore, these results indicate that the Sox2+ proliferative cells are located mainly in the PPa4 region.

The principle sources of neurons in the forebrain of adult mammals are GFAP-expressing progenitors [39]. To determine whether the Zrf1^+^ cells in the POA were proliferative, we first assayed markers of cell proliferation. Because the anti-PCNA antibody can detect distinct populations of cells, S-phase positive and S-phase non-positive [40], we compared PCNA labeling and anti-BrdU labeling in fish that were treated with BrdU (group 1 = 1 pulse day 1, three fish analyzed per group; group 2 = two pulses of BrdU at days 1 and 7; three fish analyzed per group, Figure 5). In fish treated with one pulse 1% of cells were labeled with BrdU (Figure 5A–E, green, K), about 41% of the PCNA+ cells (Figure 5K, red), 58% with BrdU+/PCNA+ (Figure 5A–E,K, yellow). In fish treated with two pulses of BrdU, 25% of the cells were BrdU+ (Figure 5F–J,M, green), 31% of the cells were PCNA+ cells (Figure 5F–J,M, red), and 44% were BrdU^+^/PCNA+ (Figure 5F–J,M, yellow, N). The number of cells, (Figure 5K,M), were analyzed according to specific regions of the hypothalamus (PPa1-PPa4, PPp1; Figure 5L,N). Thus, the labeling pattern does differ between anti-PCNA and anti-BrdU, and these data confirm that the POA generates new cells during this 7 day interval.

To determine whether the radial-glia like Zrf1+ cells were proliferative (BrdU+), we double-labeled for these markers and quantified the number of BrdU+/Zrf1+ cells. The patterns of proliferation in the POA region of the adult fish were quantified by two-pulse BrdU tracking (days 1 and 7; three fish per condition). About 74% of the total labeled cells were BrdU+/Zrf1- (Figure 6A–G, green), 23% were BrdU-/Zrf1+ (Figure 6A–G, red) and only 3% were BrdU+/Zrf1+ (Figure 6B, magnification image of boxed, arrows, G, yellow). In analyzing the distribution of the labeled cells (Figure 6H) the BrdU+/Zrf1+ cells were located primarily in the PPa2 (Figure 6H, yellow). These results suggest that most proliferative cells are not Zrf1+ (GFAP^+^), a finding different from what is observed in mammals.

### 2.4. GnRH and Testosterone Treatments Cause Neurogenesis in the POA

Previously, we have shown that GnRH or testosterone, when added to neurospheres cultured from the adult hypothalamus, can trigger differentiation of neurons in vitro [29]. To test the effects of GnRH and testosterone on the neurogenesis of the POA in vivo, adult fish were injected intraperitoneally with GnRH or testosterone, and changes in the number of BrdU+ and cytoplasmic Sox2+ cells were scored. A dose-response curve was generated to determine optimal concentrations: 15 µL per gram of 1 µM GnRH and 1 mg/mL testosterone (Appendix A). In animals treated with testosterone, we observed a slight increase in the number of BrdU+ positive cells (Figure 7A’–E’, red) relative to controls (Figure 7A–E, red). In controls the BrdU+ cells were distributed mainly along the lining of the DiV in PPa4 (Figure 7D), a pattern that was conserved in testosterone-treated animals (Figure 7D’). Analysis of the overall number of BrdU+ cells in the testosterone-treated animals showed a slight, but insignificant, increase in the number of BrdU+ cells, (Figure 7F,H; testosterone-treated: 349 ± 8 cells, eight fish; control: 232.7 ± 58 cells, 10 fish). Because the cytoplasmic Sox2+ cells represent potential neuroendocrine cells, we also quantified their response to testosterone. No significant differences were found when comparing testosterone-treated with control animals (Figure 7G,I; testosterone-treated: 76.3 ± 16.2 cells, seven fish; control: 67.2 ± 11.9 cells, six fish).

In animals treated with GnRH (Figure 8), we observed a more than 2-fold increase in the number of BrdU+ positive cells (Figure 8A’–E’, red) relative to controls (Figure 8A–E, red). Like in controls for testosterone-treated animals, in controls the BrdU+ cells were distributed mainly along the lining of the DiV in PPa4 (Figure 8D). In contrast to testosterone treatment, in fish treated with GnRH, BrdU+ cells were observed in nuclei adjacent to the DiV (Figure 8D’). Changes in the number of cytoplasmic Sox2+ cells were observed in the GnRH-treated animals (Figure 8F,F’), whereas, in the PPp1 region of the POA there was increased cytoplasmic Sox2+ cells (Figure 8F’, arrowheads, J). Quantification of the proliferation in GnRH-treated animals showed a significant increase in BrdU+ cells in GnRH-treated animals (Figure 8G; 409.7 ± 60.6 cells, nine fish; control: 173.8 ± 38.1 cells, nine fish). Significant increases in BrdU+ cells were observed in sections PPa1, PPa3, and PPa4 (Figure 8I).

Again, in contrast to treatment with testosterone, the number of cytoplasmic Sox2+ cells increased significantly in GnRH-treated animals (Figure 8H; 176.6 ± 26.7 cells, eight fish; control: 81.7 ± 16.1 cells, nine fish). Increases in cytoplasmic Sox2+ cells were observed in PPa4 and PPp1 with only the changes in PPp1 showing significance (Figure 8J). Thus, GnRH treatment promoted the proliferation of precursors (Figure 8E) and presumptive neuroendocrine cells, primarily in the PPp1 of the POA.

## 3. Discussion

Here we characterized progenitors located in the anterior/posterior parvocellular nucleus PPa/PPp of the POA (Figure 9). Within this highly proliferative area, the co-localization of cytoplasmic Sox2+ with Fezf2:GFP suggests that a subpopulation of the progenitors may be committed to a neuroendocrine fate. Furthermore, this region is responsive to hormone treatment, specifically GnRH, where in vivo GnRH treatment resulted in increased cell division and an increase in cytoplasmic Sox2+ cells. In contrast, testosterone treatment did not significantly affect the proliferative activity within this region of the hypothalamus. These results reveal for the first time a neurogenic effect of GnRH treatment in the hypothalamus of adult zebrafish in vivo. 

### 3.1. Tanycytes as Adult Neural Progenitors

In mammals, hypothalamic tanycytes characterized by the expression of progenitor cell markers, including vimentin, nestin, and Sox2 (Figure 9A, green, red; [17,34,41] with GFAP found in neural progenitors of the mammalian forebrain [39], and in α2-type tanycytes neural progenitors in the hypothalamus (Figure 9A yellow; [17]). We identified the expression of vimentin and GFAP in the dorsal regions lining the DiV and these cells had radial glia-like morphology (Figure 9B, yellow). However, the Zrf1+ cells we observed had limited proliferative capacity.

While in zebrafish, radial glia-like progenitor cells (GFAP positive) in the adult brain are thought to be the predominant neurogenic cell types [8,9,44], GFAP-negative progenitor populations have been identified in the ventral telencephalon [45]. In rats [46] and zebrafish [7], neural progenitor proliferation has been confirmed in the POA through BrdU labeling in adult animals. Here we identified Sox2+ progenitors distributed along with the DiV, as observed in the TelV of adult zebrafish [47] and in the DiV of mammals [48]. With BrdU labeling, we identified a group of proliferative cells with fusiform nuclei lining the ventral DiV (Figure 9B, grey) that are similar to neuroepithelial-like progenitors previously described by electron microscopy in the ventral region of the PPa [49] and as Nestin:GFP expressing (but not vimentin and GFAP) in the ventral telencephalon [45]. Neuroepithelial (NE) progenitor cells express progenitor markers Sox2 and nestin [9] and play a role in neural regeneration in the cerebellum of adult zebrafish [50]. Our results agree with the observation that radial glia-like progenitor cells have low proliferation (Figure 9B, yellow), and the neuroepithelial cells lining the ventral DiV are the principal proliferative population in the PP of the POA.

### 3.2. Cytoplasmic Sox2 Cells

The cell-fate determining transcription factor SOX2 plays an important role in development, stem cell biology, and cancer [51], and thus, undergoes situation-specific protein modifications that can affect the nuclear-cytoplasmic localization of the protein [52]. We found cytoplasmic Sox2+ cells in the neurosecretory preoptic area (NPO), a region with peptidergic neurons, such as those containing vasotocin and isotocin [53,54]. These cytoplasmic Sox2+ cells were Fezf2:GFP+, a transcription factor required for the development of neuroendocrine neurons in the POA [38,55,56]; thus, we proposed that cytoplasmic Sox2 cells are neurosecretory neurons. 

Interestingly, our results showed that GnRH triggered a significant increase in cytoplasmic Sox2+ cells in the PPp1, yet the significant increases in BrdU+ cells were in the adjoining PPa4 region. Both migration and transdifferentiation could explain this difference. For example, in zebrafish, the dopaminergic TH+ cells in the PPp migrate from their site of origin in the ependymo-radial glia of the anterior ventricle; thus, the increase in Sox2+ cells could be the result of migration [57]. Alternatively, the transdifferentiation of support cells into hair cells in the zebrafish ear is Sox2-dependent occurring in the absence of mitotic division [58]. Further experiments are needed to better understand the mechanisms of GnRH-induced proliferation in the POA of the adult brain.

### 3.3. Hormone Treatment and Neurogenesis

Here we have shown that GnRH and not testosterone significantly increased cell proliferation in the POA of the adult zebrafish. These results contrast with studies in the other teleosts (Tilapia), where testosterone treatment increased proliferation in the periventricular regions of the brain. However, like the findings presented here, the BrdU labeling was not found in radial glia cells [59]. Although we found a slight increase in proliferation with testosterone treatment, consistent with our previous study performed in hypothalamic neural progenitor cells in vitro, the effect was minimal with a small, but significant, increase in neurons in cultured hypothalamic progenitor cells [59]. Similarly, administration of testosterone and its metabolites 5α-dihydrotestosterone (DHT) has no effect on neurogenesis in the hypothalamus (VMH) of meadow voles [60] or hamsters [61]. Thus, to date, consistent with other studies, there is a limited neurogenic effect of testosterone on the hypothalamus of zebrafish.

### 3.4. GnRH, Neurogenesis, and Longevity

Consistent with our previous studies where GnRH had a significant effect on the differentiation of cultured hypothalamic progenitor cells from adult zebrafish [29], the results presented here showed that GnRH increased proliferation in general, and in neuroendocrine cells types. GnRH ligands and receptors are found throughout the brain in all vertebrates, and they are essential in both central and peripheral reproductive regulation, as well as higher functions, such as learning and memory, feeding behavior [62], and sleep/circadian rhythms [18]. In mice, aging is correlated with a decline in hypothalamic GnRH expression where activated IKK-β and NF-κB inflammatory crosstalk between microglia and neurons significantly down-regulates GnRH transcription [27,28]. Additionally, machine learning for predicting lifespan-extending compounds has identified compounds for GnRH therapies as one of the principal pathways for postponing the onset of many age-related diseases [63]. Hypothalamic neurogenesis, as well as other age-related phenotypes, can be restored by injection of GnRH [28,64]. Our findings that GnRH can increase proliferation of cells in the POA of adult animals, coupled with the development of zebrafish as a model for neurodegeneration in the context of aging [65], and aging-related alterations in patterns of sleep and rhythmicity patterns [66], open the door for investigations into how inhibition of inflammation and/or GnRH therapy could revert symptoms of aging-related diseases.

## 4. Materials and Methods

### 4.1. Animals

Wild-type (WT) fish of the Cornell strain (derived from Oregon AB) were raised and maintained in Whitlock laboratory in a re-circulating system (Aquatic Habitats Inc., Apopka, FL, USA) at 28 °C under a light-dark cycle of 14 and 10 h, respectively. All protocols and procedures employed were reviewed and approved by the Institutional Committee of Bioethics for Research with Experimental Animals, University of Valparaiso (#BA084-2016). The *Tg(fezf2:gfp)* line was kindly provided by Su Guo [38].

### 4.2. Histochemistry

#### 4.2.1. Trichromic Stain in Paraffin Sections

Male zebrafish, 1–2 years old, were sacrificed, heads collected, and fixed in Bouin’s solution for 24 h at 4 °C then decalcified in 0.2 Molar EDTA solution pH 7.6 for 7 days at 4 °C. Heads were rinsed in 70% ethanol, dehydrated in an increasing ethanol series to 95% ethanol, cleared in butanol, and embedded in Paraplast Plus (Sigma Chemical Co., St. Louis, MO, USA). Serial sections (5 μm) of the POA were obtained with Leica RM 2155 microtome, mounted on slides, de-paraffinized, and rehydrated. Sections processed for histology were stained with a trichromic stain (Hematoxylin/Erythrosine B-Orange G/Methyl blue) (Sigma Chemical Co., USA). The sections were then dehydrated, and mounted with Entellan (107961-Merck Millipore, Burlington, MA, USA).

#### 4.2.2. Immunocytochemistry in Paraffin Sections

Male zebrafish 1–2 years old were processed as above (trichromic stain). Serial sections 5 or 10 μm of the POA were mounted on poly-l-Lysine (Sigma) coated slides. The sections were de-paraffinized, rehydrated, and incubated in citric acid pH 6 for 30 min at 90 °C for antigen retrieval. To visualize cytoplasmic Sox2, antigen retrieval was not performed.

#### 4.2.3. Immunocytochemistry in Cryosections

Brains processed for cryostat sectioning were collected at 10 a.m. and fixed in 4% paraformaldehyde (PFA 4%) for 24 h at 4 °C. The heads were decalcified in 0.2 Molar EDTA solution pH 7.6 for 48 h at 4 °C and embedded in 1.5% agarose/5% sucrose blocks, and submerged in 30% sucrose overnight at 4 °C. Blocks were frozen at −20 °C with O.C.T. Compound (Tissue Tek^®^, Sakura Finetek, Torrance, CA, USA). Twenty μm sections were then cut using a cryostat.

#### 4.2.4. Antibodies

Information on primary antibodies is summarized in Table 1. Sections were incubated in primary antibody overnight at 4 °C and visualized using Alexa-labeled secondary antibodies (1:500; Invitrogen, Carlsbad, CA, USA). The nuclei were stained with DAPI (1:1000; Invitrogen). Sections stained for BrdU were pretreated with 2 M HCl for 15 min at 37 °C and washed with 1M PO4_4_, and incubated in BrdU (see Table 1). The labeling was visualized using Alexa-labeled secondary antibodies (1:250, Invitrogen).

#### 4.2.5. Hormone Injection and BrdU Incubation

The intraperitoneal injection procedure used was modified from [72]. The day before the experiment (day 0), male adult zebrafish (1–2 years old) were separated from females. On days 1, 3, and 7, they were anesthetized with tricaine (0.168 mg/mL) (A5040 Sigma-Aldrich, St. Louis, MO, USA), immobilized using a sponge, and injected intraperitoneally using a #70 Hamilton syringe 15 µL per gram of 1 µM GnRH (Sigma-Aldrich; L4897) diluted in saline solution, or 1 mg/mL testosterone (Sigma-Aldrich; T1500), diluted in 30% methanol and 70% saline solutions. Controls were injected with the solution used to dilute each hormone. On days 3 and 7, fish were placed in water containing 10 mM BrdU (B5002 Sigma-Aldrich) for 15 h at 28 °C, as previously described [73]. On day 8, the fish were sacrificed for immunocytochemistry.

### 4.3. Microscopy

Light field photomicrographs were taken using a Leitz-Leica DMRBE microscope (Wetzlar, Germany) equipped with a DFC290 digital camera. Fluorescent images were obtained using an Olympus BX-DSU Spinning Disc microscope (Olympus Corporation, Shinjuku-ku, Tokyo, Japan) equipped with ORCA IR2 Hamamatsu camera (Hamamatsu Photonics, Higashi-ku, Hamamatsu City, Japan) and Olympus Cell-R software (Olympus Soft Imaging Solutions, Munchen, Germany). A stack of 0.5 µm thick was collected. The images were processed using the deconvolution software AutoQuantX 2.2.2 (Media Cybernetics, Bethesda, Maryland, MD, USA) and ImageJ^®^ software (National Institute of Health, Bethesda, Maryland, MD, USA). Images are shown in Figure 4D–F) were acquired using a Nikon confocal microscope, Eclipse 80i, and analyzed with the EZ-C1 program version 3.90 NIKON.

### 4.4. Statistical Analyses

Quantification of BrdU and cytoplasmic Sox2 cells was done using Shapiro–Wilk test.The normal distribution of the data was checked under every condition. Statistical significance was evaluated by paired *t* test. Statistical analyses and graphs were done using GraphPad Prism Version 4.0 software (GraphPad Software, San Diego, CA, USA). All data were graphed with Standard Error of the Mean (SEM).

## Figures and Tables

**Figure 1 ijms-22-05926-f001:**
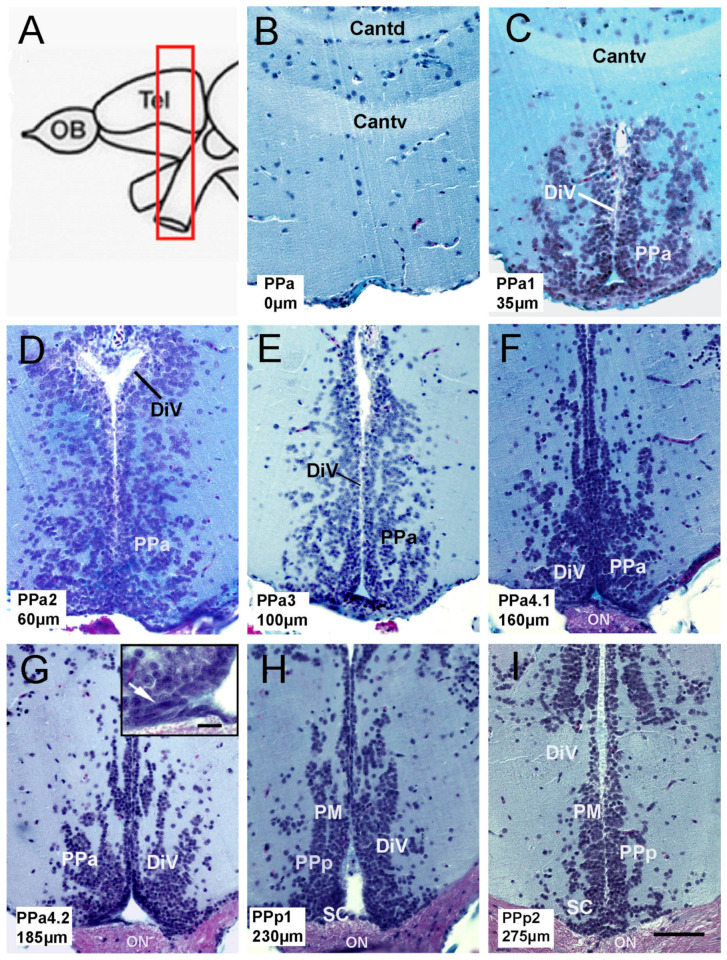
Anatomy of the preoptic area (POA) in adult zebrafish. (**A**) Schematic representation of zebrafish brain. The red, boxed area indicates the POA region. (**B**–**I**) Transverse paraffin sections of the POA stained using trichrome stain. POA region starts from the commis-sure anterior to the optic chiasm. Each slice corresponds to the representative region of the POA, categorized with a code: PPa0, PPa1, PPa2, PPa3, PPa4.1, PPa4.2, PPp1, and PPp2, where the numbers indicate the distance in μm from the beginning of the POA. (**G**) Cells with a fusiform nucleus are observed in the ventral region of the PPa (arrow in the boxed area). Commissure, pars dorsalis (Cantd); anterior commissure, pars ventralis (Cantv); diencephalic ventricle (DiV); anterior part of the parvocellular preoptic nucleus (PPa); posterior part of the parvocellular nucleus (PPp); optic nerve (ON); suprachiasmatic nucleus (SC); magnocellular nucleus (PM). Scale bar: (**I**) 50 μm, (**G**) in r.ectang.le 10 μm.

**Figure 2 ijms-22-05926-f002:**
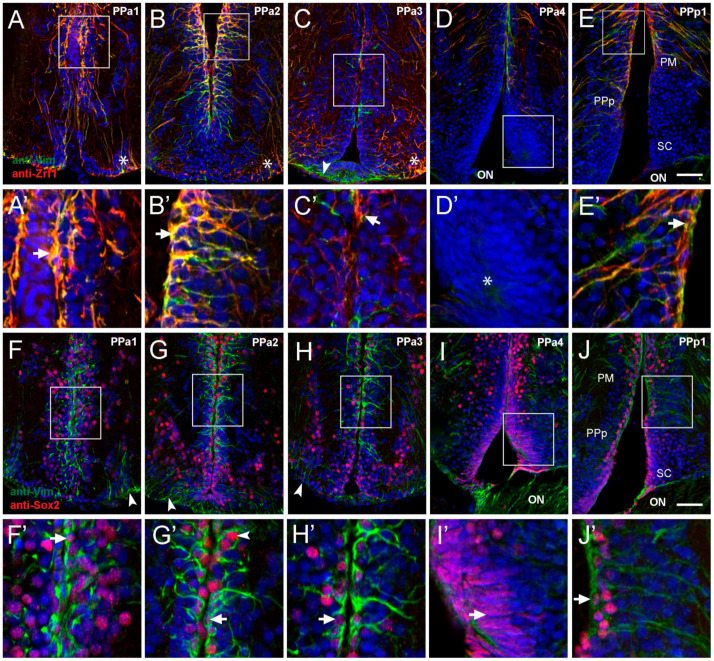
Cells lining the diencephalic ventricle (DiV) express neural progenitor markers. (**A**–**J**) Expression of Vim, Zrf1, and Sox2 in transverse paraffin-sections of 10 μm of the POA. (**A**–**E**) Cells immuno-positive for anti-Vim (red) and anti-Zrf1 (green). (**A**,**B**) Vim+/Zrf1+ positive cells are distributed dorsomedially in the wall of the third ventricle (boxed area), and signals are co-localized ((**A’**,**B’**), arrows). (**C**) Fewer Vim+/Zrf1+ cells are observed in PPa3 (boxed area) with limited co-localization ((**C’**), arrow). ((**A**–**C**), asterisks) Vim+/Zrf1+ cells are located in the ventrolateral region with long processes that extend towards the DiV (arrowheads). (**D**) Vim-/Zrf1- cells with fusiform nuclei are found in at ventral sur-face of POA ((**D’**), asterisk). (**E**) Vim+ and Zrf1+ label in dorsomedial region (boxed area), is co-localized ((**E’**), arrow). (**F**–**J**) Cells immune-positive for anti-Vim and anti-Sox2. (**F**–**H**) Vim+ cells also label with anti-Sox2 (red) (boxed area), and Sox2+ cells are observed in dorsal-ventral regions of the ventricle wall. (**F’**–**H’**) Higher magnification views of box areas in F-H: Vim+/Sox2+ co-localization in cells (arrows). (**I**) Cells with fusiform nuclei are Sox2+ and are located in the ventral wall ((**I’**), arrow). (**J**) Vim+ and Sox2+ boxed area, co-localize in ventricle wall ((**J’**), arrow). Diencephalic ventricle (DiV); magnocellular nucleus (PM); optic nerve (ON); anterior part of the parvocellular preoptic nucleus (PPa); posterior part parvocellular nucleus (PPp); suprachiasmatic nucleus (SC). All sections labeled with DAPI (blue). Scale bar: 30 μm.

**Figure 3 ijms-22-05926-f003:**
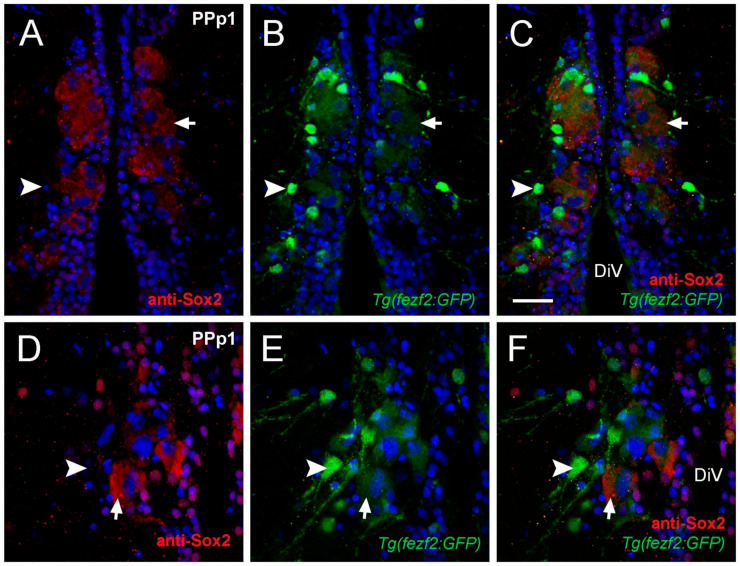
Cytoplasmic Sox2 cells co-localize with Fezf2:GFP. (**A**–**F**) Transverse cryosections of 20 µm of PPp1. (**A**–**C**) Large cytoplasmic Sox2+ cells express Fezf2:GFP (arrow), smaller cells Fezf2:GFP do not co-localize with cytoplasmic Sox2 (arrowheads). (A) Anti-Sox2 labeling. (**B**) Fezf2:GFP expression. (**C**) Merged image. Confocal images of (**D**) Sox2+ cells (**E**) Fezf2:GFP+ cells, (**F**) Merge. Cytoplasmic Sox2+ cells (red, arrow) express Fezf2:GFP (green, arrow), Small Fezf2:GFP+ cells (green, arrowhead) are Sox2 ((**D**,**F**) arrowheads). Sections labeled with DAPI (blue). Diencephalic ventricle (DiV). Scale bar: 30 μm.

**Figure 4 ijms-22-05926-f004:**
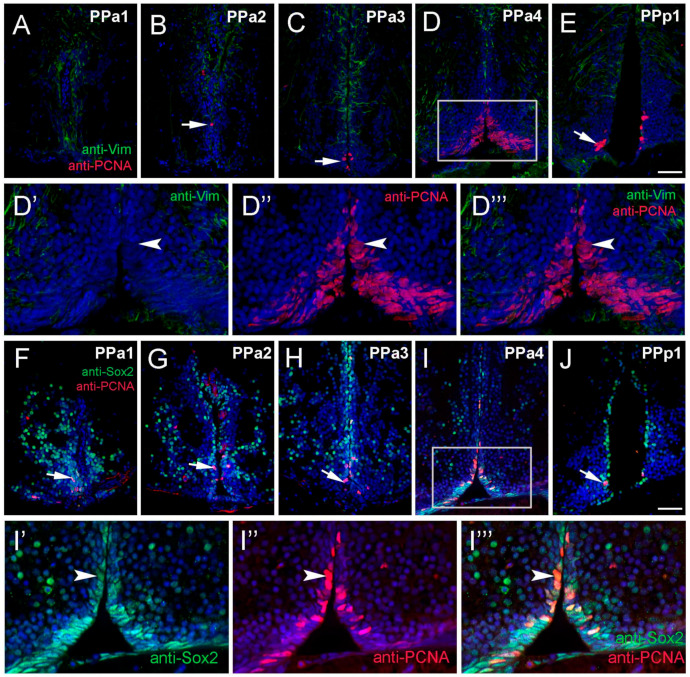
Proliferating cell nuclear antigen (PCNA) is located primarily in the PPa4/PPp1 transition region. (**A**–**J**) Transverse paraffin-sections of 5 µm of POA. (**A**–**E**) Vim+ cells (green) are localized to the dorsal region of the POA, in contrast, PCNA+ cells (red, arrow (**B**,**C**,**E**)) are found in the ventral of the POA. Vim+ labeling ((**D’**,**D’’’**), green) does not co-localize with anti-PCNA labeling ((**D’’**,**D’’’**), red). (**F**–**H**) Many Sox2+ cells (green) were observed in PPa1-3 and PPp1 (**J**) with few PCNA+ cells (red, arrows). (**I**) Sox2+ cells (green) and PCNA+ (red) in the ventral PPa4 region of the POA (boxed area) where the signals were co-localized in some cells ((**I’**–**I’’’**), arrowheads). All sections were labeled with DAPI (blue). Scale bar: 30 μm.

**Figure 5 ijms-22-05926-f005:**
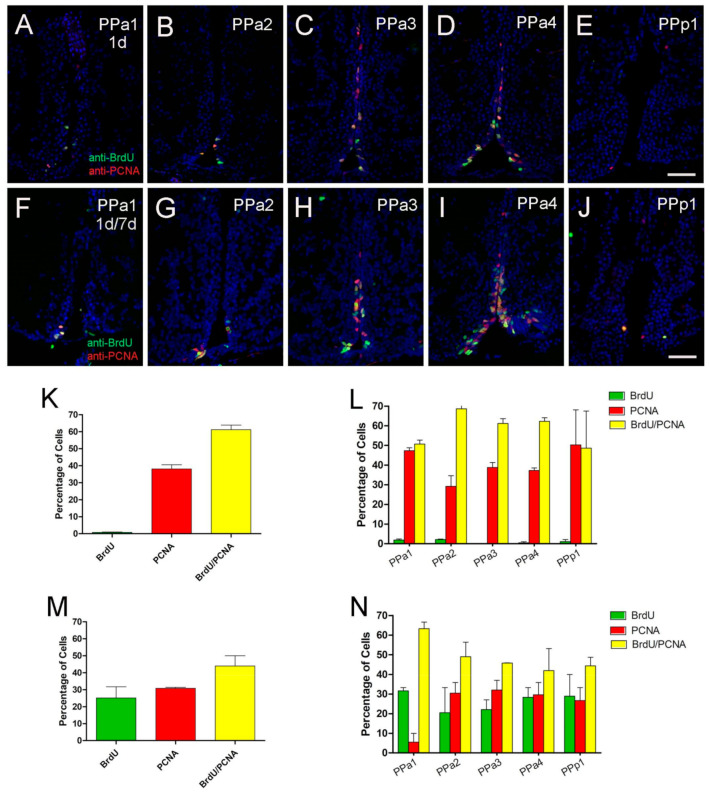
POA generates new cells with 8-day pulse-chase labeling. (**A**–**J**) BrdU/PCNA double immunofluorescence in transverse paraffin sections of 5 µm. (**A**–**E**) Treatment with one pulse of BrdU on day 1 and brains fixed on the second day. (**F**–**J**). Treatment with two pulses of BrdU at 1 and 7 days, and brains were fixed at day 8. All sections were labeled with DAPI (blue). Scale bar: 30 μm. (**K**,**M**) Percentage of cells double or single labeled for BrdU or PCNA located in the POA, with single-BrdU pulse on day 1 (**K**) and two BrdU pulses at 1 and 7 days (**M**). (**L**,**N**) Percentage of cells double or single labeled for BrdU or PCNA in representative sections of the POA with single-BrdU pulse on day 1 (**L**) and two BrdU pulses at 1 and 7 days (**N**). For each representative region of the POA, the number of sections (**L**) (PPa1 = 5, PPa2 = 10, PPa3 = 13, PPa4 = 12, and PPp1 = 4) and (**N**) (PPa1 = 5, PPa2 = 10, PPa3 = 13, PPa4 = 12, and PPp1 = 4) labeled, BrdU^+^, PCNA^+^ and BrdU^+^/PCNA^+^ cells were counted. Graphs plotted with SEM.

**Figure 6 ijms-22-05926-f006:**
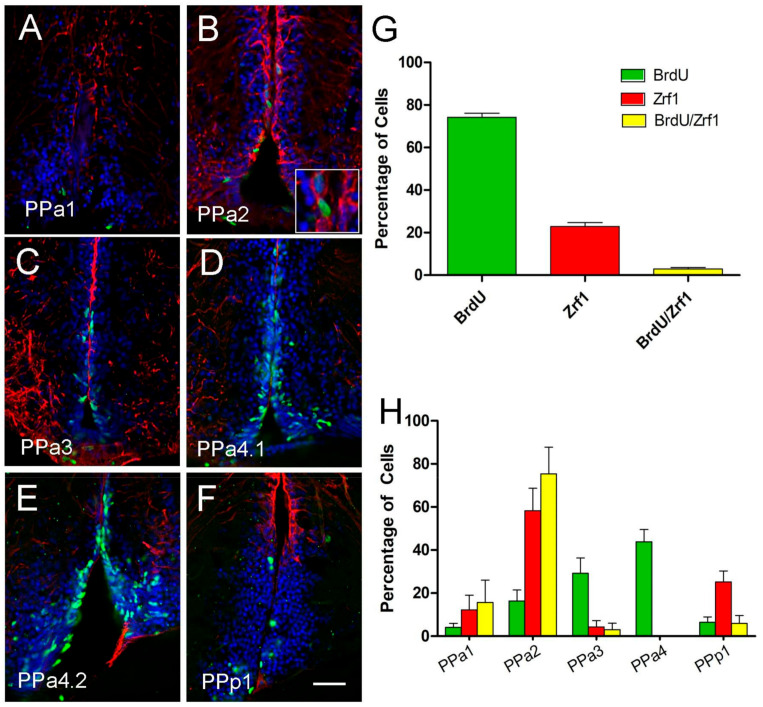
The majority of Zrf1 positive cells are non-proliferative and do not incorporate BrdU. (**A**–**F**) anti-BrdU/anti-Zrf1 double immunofluorescence in transverse cryosection of 20 µm of the POA. Treatment with two pulses of BrdU at 1 and 7 days, and the brains were fixed at 8 days. Zrf1+ tanycytes (red), generally do not express BrdU (green), but a low number of Zrf1+ cells are BrdU+ ((**B**), magnified image of boxed). All sections were labeled with DAPI (blue). Scale bar: 30 μm. (**G**) Percentage of BrdU+, Zrf1+, and Zrf1+/BrdU+ cells distributed in the wall of the DiV of the POA. The cell count was obtained of the total immune-labeling cells of the POA. (**H**) Percentage of BrdU+, Zrf1+, and BrdU+/Zrf1+ cells are located in representative sections of the POA (PPa1 = 2, PPa2 = 2, PPa3 = 2, PPa4 = 3 and PPp1 = 2) where BrdU+, Zrf1+, and Zrf1+/BrdU+ cells were counted.

**Figure 7 ijms-22-05926-f007:**
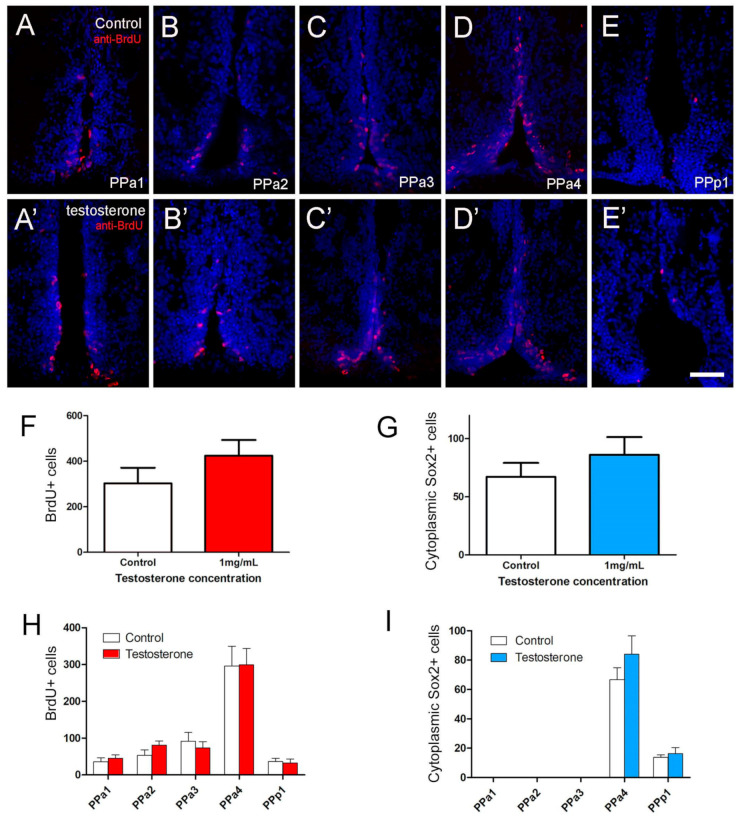
Testosterone did not affect cell proliferation in the POA. (**A**–**E**,**A’**–**E’**) Quantification of the effects of testosterone (1 mg/mL) in POA of adult zebrafish. The number of cells was counted in transverse 20 µm cryosections of the POA (12 sections). (**A**–**E**) Testosterone control. (**A’**–**E’**) Testosterone treatment (1 mg/mL). (**F**) Number of BrdU+ cells. (**G**) Number of cytoplasmic Sox2+ cells. (**H**,**I**) Number of cells located in the regions of the POA (20 µm: PPa1 = 2, PPa2 = 2, PPa3 = 3, PPa4 = 4 and PPp1 = 2). All sections were labeled with DAPI (blue). Scale bar: 30 μm.

**Figure 8 ijms-22-05926-f008:**
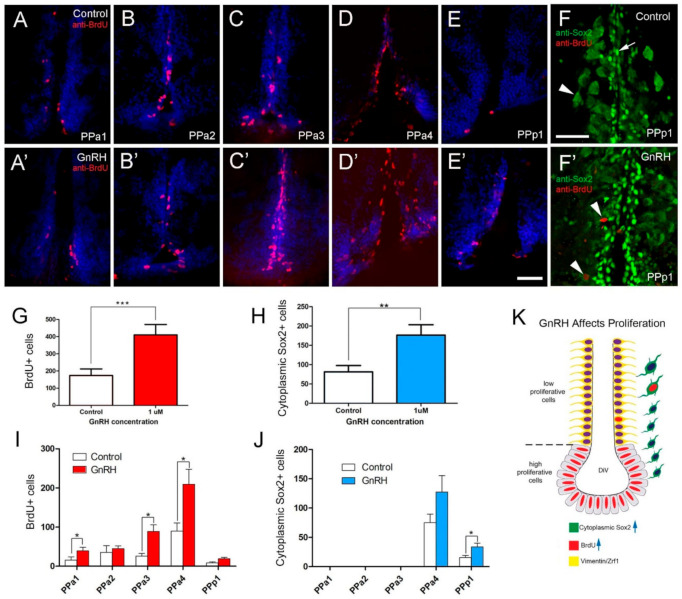
BrdU labeling reveals GnRH-induced increases in cells in the POA. (**A**–**E**,**A’**–**E’**) Quantification of the effects of GnRH (1 µM GnRH) on the POA of adult zebrafish: Cells were counted in transverse 20 µm cryosections of the POA (12 sections). (**A**–**E**) GnRH control. (**A**–**E’**) GnRH treatment (1 µM GnRH). (**F**) Control with Sox2+ cells (arrow), cytoplasmic Sox2+ (arrowhead) (**F’**) GnRH treatment increased cytoplasmic Sox2+ cells in the PPp1 (arrowheads). (**G**) Significant increases in BrdU labeled cells in the GnRH-treated animals. (**H**) Significant increases in cytoplasmic Sox2+ cells increased in the GnRH-treated animals. *** *p* < 0.001; ** *p* < 0.01; Student’s *t*-test, SEM. (**I**,**J**) Number of cells located in the regions of the POA scored in transverse cryosections (20 µm: PPa1 = 2, PPa2 = 2, PPa3 = 3, PPa4 = 4 and PPp1 = 2). (**I**) Number of BrdU+ cells (**J**) number of cytoplasmic Sox2+ cells in GnRH-treated animals. * *p* < 0.05, Student’s *t*-test, SEM. (**K**) GnRH treatment increased the number of BrdU+ cells (blue arrow) and cytoplasmic Sox2+ (blue arrow) cells, where few cytoplasmic Sox2+ were also BrdU+. (**A**–**E**,**A’**–**E’**): Sections labeled with DAPI (blue). Scale bars: 30 μm.

**Figure 9 ijms-22-05926-f009:**
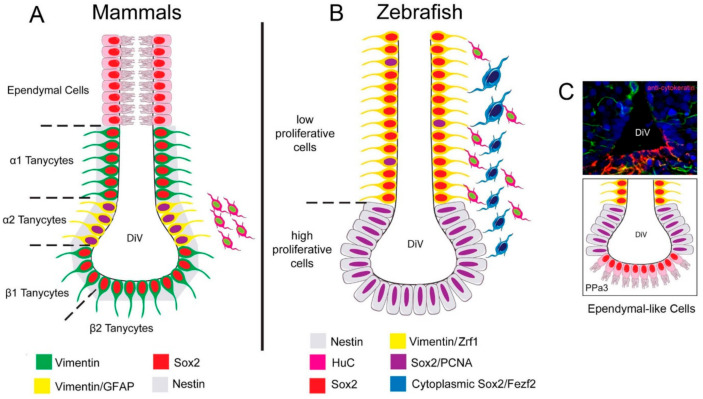
Summary of the distribution of neural progenitors in the POA. (**A**) Distribution of neural progenitors lining the DiV in the hypothalamus of mammals according to [17]. α2 tanycytes with proliferative capacity express: Vim, Nestin [34], Sox2, and GFAP, while the non-proliferative tanycytes do not express GFAP. Ependymal cells are distributed in the dorsal region. (**B**) In zebrafish, we described a low-proliferative cell Vim+, Sox2+, and Zrf1+, and high proliferative Sox2+ cells, previously reported to express nestin [42]. Cytoplasmic Sox2/Fezf2:GFP cells were observed in the region previously shown to express cytoplasmic Sox2 [29]. (**C**) Cytokeratin cells are express in the ventral region of the POA in cytokeratin [43] positive ependymal-like cells similar to those seen in mammals.

**Table 1 ijms-22-05926-t001:** Primary antibodies that are used for immunocytochemistry.

Antigen	Host	Dilution	Manufacturer	Cat. No	Immunogen Organism	Reference Use in Zebrafish
BrdU	Rabbit	1:500	Invitrogen	PA5-32256	BrdU conjugated to KLH	Not identified
BrdU	Rat	1:500	Abcam	Ab6326	Not available	[67]
GFP	Mouse	1:500	Invitrogen	A-11120	*Aequorea victoria*	[68]
PCNA	Mouse	1:100	Sigma-Aldrich	P8825	Rat	[69]
Sox2	Mouse	1:200	Abcam	ab137385	Human	Not identified
Sox2	Rabbit	1:500	Abcam	Ab97959	Human	[70]
Vimentin	Chicken	1:500	Millipore	AB5733	Human	[29]
Zrf1	Mouse	1:500	Abcam	ab154474	Zebrafish	[71]
Cytokeratin(clones AE1/AE3)	Mouse	1:500	Dako	M3515	Human	[43]

## Data Availability

The datasets used and/or analyzed during the current study are available from the corresponding author on reasonable request.

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
