# Peer review of "Gonadotropin Releasing Hormone (GnRH) Triggers Neurogenesis in the Hypothalamus of Adult Zebrafish"

_ijms, 2021, doi:10.3390/ijms22115926_

Round 1

Reviewer 1 Report

Authors characterized tanycyte like cells of adult zebrafish preoptic area to determine their neurogenic potential. They found Sox2 expression both in Vim/GFAP+ subpopulation but also in Vim/GFAP- fusiform cells lining the ventral part of diencephalic ventricle. In addition, cytoplasmic Sox2 expression was also detected in Fezf2-positive neurons of the neurosecretory preoptic area. However, most of Vim/GFAP positive cells are low proliferating, whereas high proliferating cells, including ventral fusiform Sox2 positive cells, are not characterized. Which kind of cells are these since they are Vim/GFAP negative? Then, authors assessed the effect of GnRH and testosterone treatments on hypothalamic neurogenesis and found that GnRH increases cell proliferation and the number of Sox2/Fezf2 positive cells. However, cytoplasmic sox2-positive cell increase is seen in PPp1 area whereas Brdu+ cells increase is observed in PPa areas: authors should add an explanation in the discussion.

Fig. 2 Additional high mag images of ventrolateral cells in A, B and C panels would be helpful to identify Vim+/Zrf1- cells and double positive cells; similarly, for F,G and H panels high mag of Vim+/sox2- ventrolateral cells. In presented high mag images arrows point at double positive cells in the DiV wall while in low mag arrows are used to highlight Vim+/Zrf1- cells and double positive cells are highlighted by arrowheads. Please uniform this: e.g. arrows for colocalization in both high and low mag and arrowheads for lack of colocalization

Fig. 2E differences between processes shown in A-B and E panels are not so clear; authors may consider to add also a schematic drawing of coronal sections of relevant areas would be helpful for readers not familiar with zebrafish anatomy 

Paragraph 2.3: wrong references to figure 4 and figure 6 are present in the text

Fig.7-8 I suggest to re-arrange figures including quantification of Brdu+ cells in fig.7. Now quantification are reported in fig.8A-C (no references in the text) for GnRH treatment and in suppl fig.3A-C (no references in the text) for testosterone treatment. Control columns are coloured in some graphs and white in others. Images of relevant sections showing increase cytoplasmic sox2 cells are missing and co-localization between cytoplasmic Sox2 and Brdu might be provided.

Fig.9 lacking A and B; colors legend is not completely represented in the images (e.g. nestin in mammals schematics)

Error bars refer to SD or SEM?

Author Response

We are returning our revised manuscript entitled: Gonadotropin releasing hormone (GnRH) triggers neurogenesis in the hypothalamus of adult zebrafish, by Ricardo Ceriani and Kathleen E Whitlock *.

In response to the review by experts in the field and subsequent request that we make major revisions before it is processed further, we have responded to the comments as outlined below. We have made extensive changes to the manuscript. Including changes that were not requested by the reviewers, to correct errors in the text and call to figures.

In regard to the Figures the following have been changed:

Figure 1: Small correction in image panel 1E

Figure 2: Corrections in labeling in response to the reviewers

Figure 3: No revisions

Figure 4: Small correction in labeling D’-D’’’

Figure 5: No revisions

Figure 6: No revisions

Figure 7: Major revision: In response to comments by the reviewer, this figure now includes the data on the testosterone treatment, part of which was in Supplemental Figure 3.

Figure 8: Major revision in response to the reviewers: all images and graphs corresponding to the GnRH treatment are now included in this figure. In addition (in response to the reviewers) we have added images of BrdU/Sox2 labeling (F, F’).

Figure 9: Revision in response to the reviewers to clarify the summary diagram.

Supplemental Figure 1: No revisions

Supplemental Figure 2: No revisions

Supplemental Figure 3: Eliminated

REVIEWER #1

Authors characterized tanycyte like cells of adult zebrafish preoptic area to determine their neurogenic potential. They found Sox2 expression both in Vim/GFAP+ subpopulation but also in Vim/GFAP- fusiform cells lining the ventral part of diencephalic ventricle. In addition, cytoplasmic Sox2 expression was also detected in Fezf2-positive neurons of the neurosecretory preoptic area.

However, most of Vim/GFAP positive cells are low proliferating, whereas high proliferating cells, including ventral fusiform Sox2 positive cells, are not characterized. Which kind of cells are these since they are Vim/GFAP negative?

RESPONSE: We have addressed this cell type in the dicssion, please see:

With BrdU labeling we identified a group of proliferative cells with fusiform nuclei lining the ventral DiV (Fig. 9, B, grey) that are similar to neuroepithelial-like progenitors previously described by electron microscopy in the ventral region of the PPa [43] and as expressing Nestin:GFP (but not vimentin and GFAP) in the ventral telencephalon, [39]. Neuroepithelial (NE) progenitor cells express progenitor markers Sox2 and Nestin [9] and play a role in neural regeneration in the cerebellum of adult zebrafish [44].

Then, authors assessed the effect of GnRH and testosterone treatments on hypothalamic neurogenesis and found that GnRH increases cell proliferation and the number of Sox2/Fezf2 positive cells. However, cytoplasmic sox2-positive cell increase is seen in PPp1 area whereas Brdu+ cells increase is observed in PPa areas: authors should add an explanation in the discussion.

RESPONSE: we have added the following section toi the Discussion: Interestingly, our results showed that GnRH triggered a significant increase in cytoplasmic Sox2+ cells in the PPp1, yet the significant increases in Brdu+ cells were in the adjoining PPa4 region. Both migration and transdiferentiation could explain this difference. For example in zebrafish the dopaminergic TH+ cells in the PPp migrate from their site of origin in the anterior ventricular ependymo-radial glial, thus the increase in Sox2+ cells could be the result of migration (57). Alternatively, the transdifferentiation of support cells into hair cells in the zebrafish ear is Sox2-dependent occurring in the absence of mitotic division (58). Further experiments are needed to better understand the mechanisms of GnRH induced proliferation in the POA of the adult brain.

Fig. 2 Additional high mag images of ventrolateral cells in A, B and C panels would be helpful to identify Vim+/Zrf1- cells and double positive cells; similarly, for F,G and H panels high mag of Vim+/sox2- ventrolateral cells.

RESPONSE: Unfortunately we do not have higher magnification views of these preparations: the boxed areas are images taken with 100X objective. Furthermore this moment it is not possible to enter the imaging facility to try and take more photos on the confocal at higher magnification as we are in full lockdown.

In presented high mag images arrows point at double positive cells in the DiV wall while in low mag arrows are used to highlight Vim+/Zrf1- cells and double positive cells are highlighted by arrowheads. Please uniform this: e.g. arrows for colocalization in both high and low mag and arrowheads for lack of colocalization

RESPONSE: We have revised this section correcting the labeling in Figure 2 as well as correcting the text that corresponds to this figure.

Fig. 2E differences between processes shown in A-B and E panels are not so clear; authors may consider to add also a schematic drawing of coronal sections of relevant areas would be helpful for readers not familiar with zebrafish anatomy 

RESPONSE: We are confused about this comment: The purpose of Figure1 is to orient readers not familiar with the zebrafish brain to the specific morphology of the region in the hypothalamus that we are analyzing in this study. Additionally on page 5 we cite the well-known book of Wulliman on zebrafish morphology (Cells with fusiform nuclei were no longer apparent in the posterior parvocellular preoptic nucleus (PPp) /suprachiasmatic (SC) (Fig. 1, H-I) in agreement with Wulliman [33].)

Paragraph 2.3: wrong references to figure 4 and figure 6 are present in the text.

RESPONSE: We have clarified the call to Figures in section 2.3.

Fig.7-8 I suggest to re-arrange figures including quantification of Brdu+ cells in fig.7. Now quantification are reported in fig.8A-C (no references in the text) for GnRH treatment and in suppl fig.3A-C (no references in the text) for testosterone treatment. Control columns are coloured in some graphs and white in others.

RESPONSE: Figures 7 and 8 have been rearranged: Figure 7 has the complete dataset for testosterone treatment (Supplemental Figure 3 has been eliminated and incorporated into the New Figure 7). We have corrected the coloring of the columns in the graphs.

Images of relevant sections showing increase cytoplasmic sox2 cells are missing and co-localization between cytoplasmic Sox2 and Brdu might be provided.

RESPONSE: We have added two panels to Figure 8, (F, F’) from control/ GnRH treated animals, in the region of the PPp1 where we have observed increased cytoplasmic Sox2+ cells.

Fig.9 lacking A and B; colors legend is not completely represented in the images (e.g. nestin in mammals schematics)

RESPONSE: We have added A, B, and C; references for the Nestin labeling; clarificacion for the ependymalcells/ependymal-like cells with an image.

Error bars refer to SD or SEM?

RESPONSE: We have clarifies that data are plotted with SEM

Reviewer 2 Report

Ceriani and Withtlock have submitted results demonstrating the impact of GnRH on neurogenesis in zebrafish. This work is very well done and results are convincing and bring very interesting informations on the role of GnRH.

I have few remarks on the text :

1) To characterize the cell immunogenicity avoid the term “anti-Vimentin”, use [Vim+] or [Vim-]. Avoid the formulation “examined the expression of anti-Vimentin (Vim), anti-Zrf1 (Zrf1)…”, use “examined the presence of Vim protein , Zrf1 protein…

2) You have written (page 4) : “In PPa3 sections the anti-Vim+/Zrf1+ began to disappear”. Is it a real disappearance ? Or is it a different distribution…

3) Same remark when you wrote (page 4) “In PPp1, anti-Vim+/Zrf1+ cells lining the DiV Appeared in the magnocellular”. For me it’s a question of distribution, not appearance or disappearance.

For the scientific aspect :

1) You wrote (page 4) that tanycyte-like ependymal cells express GFAP and SOX2. But you didn’t show double labelling Zrf1/Sox2. According your Mat and Meth this experiment is possible (primary antibodies table). This double labelling Zrf1/Sox2 is essential to precise the tanycyte-like phenotype. In your Mat and Meth you have indicated the use of GFAP antibody (primary antibodies table) but I didn’t see immunostaining with GFAP antibody in the manuscript.

2) Even if you said (page 6) cytoplasmic [Sox2+] are [HuC+] it would interesting to phenotype [Sox2+ ; GFP+] cells with specific neuron cells or glial cells markers. Indeed, the morphology of [Sox2+ ; GFP+] are closer to glial cell morphology than neuron cells morphology.

3) Page 10, if the question is the % of [BrDU+] cells in the [Zrf1+] population why presenting [BrDU+ ; Zrf1-] result. In this case you have to determine the % of [BrDU- ; Zrf1-].

Experiments have to be completed to answer at the remarks 1) and 2).

Author Response

We are returning our revised manuscript entitled: Gonadotropin releasing hormone (GnRH) triggers neurogenesis in the hypothalamus of adult zebrafish, by Ricardo Ceriani and Kathleen E Whitlock *.

In response to the review by experts in the field and subsequent request that we make major revisions before it is processed further, we have responded to the comments as outlined below. We have made extensive changes to the manuscript. Including changes that were not requested by the reviewers, to correct errors in the text and call to figures.

In regard to the Figures the following have been changed:

Figure 1: Small correction in image panel 1E

Figure 2: Corrections in labeling in response to the reviewers

Figure 3: No revisions

Figure 4: Small correction in labeling D’-D’’’

Figure 5: No revisions

Figure 6: No revisions

Figure 7: Major revision: In response to comments by the reviewer, this figure now includes the data on the testosterone treatment, part of which was in Supplemental Figure 3.

Figure 8: Major revision in response to the reviewers: all images and graphs corresponding to the GnRH treatment are now included in this figure. In addition (in response to the reviewers) we have added images of BrdU/Sox2 labeling (F, F’).

Figure 9: Revision in response to the reviewers to clarify the summary diagram.

Supplemental Figure 1: No revisions

Supplemental Figure 2: No revisions

Supplemental Figure 3: Eliminated

REVIEWER #2:

Ceriani and Withtlock have submitted results demonstrating the impact of GnRH on neurogenesis in zebrafish. This work is very well done and results are convincing and bring very interesting informations on the role of GnRH.

I have few remarks on the text :

1) To characterize the cell immunogenicity avoid the term “anti-Vimentin”, use [Vim+] or [Vim-]. Avoid the formulation “examined the expression of anti-Vimentin (Vim), anti-Zrf1 (Zrf1)…”, use “examined the presence of Vim protein , Zrf1 protein…

RESPONSE: We have changed the wording to the suggested Vim+ or Vim- not only in this section but have used this format throughout the manuscript.

2) You have written (page 4) : “In PPa3 sections the anti-Vim+/Zrf1+ began to disappear”. Is it a real disappearance ? Or is it a different distribution…

RESPONSE: The reviewer is correct, this is poor word usage.We have revised this section to improve the wording and eliminate words like “disappear”.

3) Same remark when you wrote (page 4) “In PPp1, anti-Vim+/Zrf1+ cells lining the DiV Appeared in the magnocellular”. For me it’s a question of distribution, not appearance or disappearance.

RESPONSE: We have revised this section to improve the wording and eliminate words like “appeared”.

For the scientific aspect :

1) You wrote (page 4) that tanycyte-like ependymal cells express GFAP and SOX2. But you didn’t show double labelling Zrf1/Sox2. According your Mat and Meth this experiment is possible (primary antibodies table). This double labelling Zrf1/Sox2 is essential to precise the tanycyte-like phenotype.

RESPONSE: The reason for the GFAP (Zrf1) double labeling with sox2 would be to show that the Zrf1 cells (a marker for tanycytes) also express a progenitor marker (Sox2). We have performed the equivalent labeling which is Zrf1 with BrdU (Figure 6).

In your Mat and Meth you have indicated the use of GFAP antibody (primary antibodies table) but I didn’t see immunostaining with GFAP antibody in the manuscript.

RESPONSE: We did not use GFAP because it is not made against the zebrafish protein. The anti-zrf1 antibody is made against the zebrafish GFAP protein. We have eliminated the part of the Table referring to the GFAP antibody.

2) Even if you said (page 6) cytoplasmic [Sox2+] are [HuC+] it would interesting to phenotype [Sox2+ ; GFP+] cells with specific neuron cells or glial cells markers. Indeed, the morphology of [Sox2+ ; GFP+] are closer to glial cell morphology than neuron cells morphology.

RESPONSE: We are a bit confused by what the reviewer means by GFP+, we assume the Fezf2:GFP line? In this manuscript we are specifically looking for tanycyte-like cells which is why we did not direct many experiments to understanding neuronal morphology and concentrated on markers that define tanycytes.

  1. A) The work with HuC we referred to is from a previous paper from our lab on neurospheres. The reference to our previous work is to show that we have localized Sox2 to neurons, but in agreement with the literature Sox2 is also in glia cells. We have added the reference :

Mercurio, S et al (2019). More than just Stem Cells: Functional Roles of the Transcription Factor Sox2 in Differentiated Glia and Neurons. Int J Mol Sci Sep; 20(18); 4540

  1. B) In this manuscript we used the Tg(fezf2:gfp) line because the expression pattern was reminiscent of the cytoplamic sox2 labeling and this has been shown to ne a neuronal marker

3) Page 10, if the question is the % of [BrDU+] cells in the [Zrf1+] population why presenting [BrDU+ ; Zrf1-] result. In this case you have to determine the % of [BrDU- ; Zrf1-].

RESPONSE: The percentages are calculated only for labeled cells, it is extremely difficult to count the labeled cells thus we did not include the cells that were only DAPI positive in the analysis. Perhaps the reviewer is referring to Fig. 6 where we show BrdU+ as a column? This is to give an idea of the overall level of BrdU labeling.

Experiments have to be completed to answer at the remarks 1) and 2).

Sincerely,

Kathleen Whitlock, Ph.D.

Instituto de Neurociencia

Round 2

Reviewer 1 Report

Authors positively reply to most of the issue raised with the exception of high magnification images requested for Figure 2: I think this would be an added value to the work but I understand difficulties caused by full lockdown.

Reviewer 2 Report

Dear authors,

Thank you for your answers. Modifications in the text reorient scientific questions. It's clearer for me.

Congratulations for your work.